# Adenovirus 36 Infection in People Living with HIV—An Epidemiological Study of Seroprevalence and Associations with Cardiovascular Risk Factors

**DOI:** 10.3390/v14081639

**Published:** 2022-07-27

**Authors:** Mariusz Sapuła, Magdalena Suchacz, Joanna Kozłowska, Aneta Cybula, Ewa Siwak, Dagny Krankowska, Alicja Wiercińska-Drapało

**Affiliations:** Department of Infectious and Tropical Diseases and Hepatology, Medical University of Warsaw, 01-201 Warsaw, Poland; magdalena.suchacz@wum.edu.pl (M.S.); askak1@o2.pl (J.K.); anetacybula@gmail.com (A.C.); siwakeb@gmail.com (E.S.); d.krankowska@gmail.com (D.K.); awiercinska@gmail.com (A.W.-D.)

**Keywords:** HIV, adenovirus 36, obesity, metabolic syndrome, cardiovascular risk

## Abstract

Background. With the life expectancy of people living with HIV (PLHIV) rapidly approaching that of the general population, cardiovascular health in this group is as relevant as ever. Adenovirus 36 (Adv36) is one of the few viruses suspected to be a causative factor in promoting obesity in humans, yet there is a lack of data on this infection in PLHIV. Methods. PLHIV on stable suppressive antiretroviral therapy were included in the study, with assessment of anthropometric measures, blood pressure, serum lipid levels, fasting serum glucose and insulin, non-classical serum cardiovascular risk markers related to inflammation (hsCRP, resistin, calprotectin), and anti-Adv36 antibodies during a routine check-up. Results. 91 participants were recruited, of which 26.4% were Adv36-seropositive (Adv36(+)). Compared to Adv36-seronegative (Adv36(−)) controls, Adv36(+) individuals had a lower waist circumference (Adv36(+) 89.6 ± 7.7 cm, Adv36(−) 95.5 ± 11.7 cm, *p* = 0.024) and a lower waist-to-hip ratio (Adv36(+) 0.88 ± 0.06, Adv36(−) 0.92 ± 0.09, *p* = 0.014), but this did not reach statistical significance in the multivariate analysis (*p* > 0.05). Adv36(+) participants were less likely to be on lipid-lowering treatment (Adv36(+) 12.5%, Adv36(−) 34.3%, *p* = 0.042), even after adjustment for relevant baseline characteristics (OR = 0.23, 95%CI = 0.04–0.91), but no differences in cholesterol or triglyceride levels were found. No other statistically significant associations were observed. Conclusions. We found no evidence to support the claim that past Adv36-infection is associated with an increased prevalence of cardiovascular risk factors or with elevated inflammatory markers in PLHIV. More research is needed to replicate these findings in other samples of PLHIV and to compare them with the HIV-negative population.

## 1. Introduction

With the life expectancy of people living with HIV rapidly approaching that of the general population [1], HIV research has understandably taken an interest in classical and non-classical cardiovascular disease risk factors in this population. Comorbidities such as diabetes, cardiovascular disease, or kidney disease have been observed to occur earlier in PLHIV than in their HIV-negative counterparts [1], and HIV infection itself is considered a cardiovascular risk factor due, among other mechanisms, to the persistent immune activation present despite suppressive antiretroviral treatment [2]. Antiretroviral drugs, on the other hand, are under continued scrutiny for their impact on metabolic health, with increased weight gain observed particularly with some integrase-strand inhibitors, and with tenofovir alafenamide and a negative impact on plasma lipid levels seen with some protease inhibitors [3].

Adenovirus 36 (Adv36) is one of the few viruses suspected to be a causative factor in promoting obesity in humans. This was first reported in 2000 by Dhurandar et al., who infected chickens and mice with this human virus, observing a marked increase in body and visceral fat and a paradoxical decrease in serum total cholesterol and triglyceride levels relative to control, non-infected chickens and mice [4]. Similar increases in adiposity after infection with Adv36 were observed in rats (increased adiposity and increased insulin sensitivity [5]), rhesus monkeys (increased adiposity and lowering of serum total cholesterol, production of anti-Adv36 antibodies [6]), marmosets (increased adiposity and lowering of serum total cholesterol [6]), and hamsters (increased adiposity and increase in serum LDL cholesterol [7]).

Infection with other viruses, including adenovirus 5 and SMAM-1, have also been linked to obesity (sometimes denominated ‘infectobesity’), but studies performed to date have yielded less of a clear association [8,9].

In epidemiological studies on human populations, Adv36 seroprevalence ranges considerably from 7% to 58% and is higher in obese patients than in non-obese patients in most reports [10,11,12,13,14,15,16,17], although not all [18,19,20,21]. In one study, there was an inverse relationship between Ad36 seropositivity and the odds of obesity in Chinese Han men (but not in women) [20], suggesting that other environmental or genetic factors may also influence the course of Adv36 infection. Four meta-analyses conducted on the topic focused on different observational studies, but all four found a roughly two-fold increase of the odds of obesity with Ad36 infection [22,23,24,25].

Human epidemiological studies have also explored the association of Adv36-seropositivity with classical cardiovascular risk factors. What was found was a generally neutral effect on serum triglyceride and cholesterol concentrations [11,14,15,16,18,20,21], with two studies reporting a beneficial association [17,19] and two studies a proatherogenic association [10,26]; increased insulin sensitivity in non-diabetic patients, as well as fasting glycaemia in both non-diabetic and diabetic patients in one study [17], although not replicated in other reports [10,11,15,16,20,21]; conflicting reports on blood pressure [10,20], with some studies showing a neutral association [17,19]. These studies were relatively heterogeneous in their design and statistical analysis, but generally showed a clinically neutral or beneficial association of Adv36 seropositivity with metabolic status. The interpretation of these results is complicated by the positive association of Adv36 seropositivity with obesity and a potential inverse association of Adv36 seropositivity with other components of the metabolic syndrome, which can lead to statistical confounding, which was not always taken into account.

The mechanism by which Adv36 infection could lead to obesity remains obscure. The results of experimental in vitro and in vivo studies have suggested that one of the mechanisms by which Adv36 can cause obesity is the stimulation of preadipocite differentiation and increase in the lipid content of the cell [8]. Egrin et al. [13] could not find Adv36 DNA in adipose tissue of obese patients undergoing liposuction (including six patients who were seropositive for Adv36), even though a higher seroprevalence of Adv36 was found in the obese than in the non-obese group, and in the obese group Adv36-positive patients had a higher BMI than Adv36-seronegative patients. This suggests that a ‘hit-and-run’ (a term first used in the context of Adv36 by Karamese et al. [14]) mechanism may be at play here, where Adv36 infection causes the expansion of adipose tissue that is not reversible with the resolution of the acute phase of the infection. This would suggest that Adv36 serology is as robust, if not more, for most studies on the long-term effects of Adv36 infection than nucleic acid detection methods. On the other hand, Waye et al. detected Adv36-DNA in plasma and in stool in roughly 7% and 14% of participants, respectively [15], and, as such, persistent or intermittent Adv36 replication can play a role in Adv36-associated weight gain as well.

## 2. Materials and Methods

In this cross-sectional study, we aimed to explore the associations between cardiovascular risk factors, inflammatory markers, and Adv36-serostatus. To this end, patients treated for HIV in the Department of Infectious and Tropical Diseases and Hepatology of the Medical University of Warsaw, Poland were asked to participate in the study during a routine HIV check-up. We included adult patients who had been on antiretroviral therapy for at least 6 months and who had HIV suppression, defined as a HIV plasma viral load <50 copies/mL. We excluded patients that had current (but not past) opportunistic disease or had a new or exacerbated medical problem at the time of the visit. Upon consent, anthropometric measurements were made by a physician and a routine blood draw was taken.

Most of the blood biomarkers were determined as a routine part of the check-up visit. Determinations of anti-Adv36-antibodies, high-sensitivity C-reactive protein (hsCRP), resistin, and calprotectin were achieved in a separate research laboratory on patients’ serum. The determination of anti-Adv36-antibodies was achieved using a qualitative ELISA, which was manufactured by Abbexa.

The study was conducted in accordance with the Declaration of Helsinki and approved by the Bioethics committee of the Medical University of Warsaw (protocol code KB/97/2021, date of approval 2 June 2021).

For data analysis, the chi-squared test (or, where more appropriate, Fisher’s exact test) and the Student’s t-test were used in a univariate analysis to compare the qualitative and quantitative variables between groups, and, subsequently, a multivariate analysis was undertaken using multiple regression analysis and multiple logistic analysis. The R statistical analysis software version 4.1.3 was used for all calculations.

## 3. Results

A total of 91 patients were recruited into the study. Baseline characteristics and a univariate comparison between Adv36-seropositive and Adv36-seronegative individuals are shown in Table 1.

In order to adjust for relevant baseline characteristics, multivariate analysis was undertaken. Comparisons adjusted by baseline characteristics through multiple regression and multiple logistic analyses are shown in Table 2 (anthropometric measures), Table 3 (quantitative cardiovascular risk factors), and Table 4 (odds of being on pressure-, lipid-, or glucose-lowering treatment depending on Adv36-serostatus).

## 4. Discussion

To our knowledge, this is the first report of the associations between Adv36-seropositivity and cardiovascular risk factors in PLHIV.

The Adv36-seroprevalence in our sample was 26.4%, which is within range of that observed worldwide and comparable to the Adv36-seroprevalence that was seen in studies conducted in countries geographically neighboring Poland, i.e., Czechia (Adv36-seroprevalence of 26.5% in adolescents [26]) and Sweden (Adv36-seroprevalence of 7–20% [11]). These studies were performed on (predominantly) HIV-negative populations.

The only data from the Polish population pertaining to this subject is a study on the prevalence of serotype-non-specific IgG against adenoviruses, and it found a prevalence of 89.5% of Ad-seropositivity in the studied sample [27]. Interestingly, there was a positive association between non-specific anti-adenoviral seropositivity and obesity. This result is difficult to compare with the results in our study, as we detected specific anti-Adv36 antibodies instead.

In the univariate analysis in our sample, Adv36-positive patients had, on average, a significantly smaller waist circumference and a significantly smaller waist-to-hip ratio, with no statistically significant differences in BMI. In the multivariate analysis, these differences were also present, but were not statistically significant at alpha = 0.05. This was somewhat surprising to us, as we expected an inverse relationship given the predominance of animal and human studies linking past Adv36-infection to obesity. Though the idea that past Adv36-infection is a protective factor for the development of obesity in PLHIV is a tempting one, we still find it an unlikely one given the plethora of experimental and epidemiological data linking Adv36-infection to proadipogenic effects.

Mirroring epidemiological trends in Poland [28], our sample was composed in over 90% of men, which could potentially influence the results we obtained and may limit the generalizability of our results. A similar negative association between Adv36-seropositivity and obesity was observed in a study by Zhou et al. in the Han population specifically in men, where Adv36-seropositivity was associated with lower odds of obesity [20]. A difference between genders was also seen in Trovato’s et al. study [10], where there was a positive association between Adv36-seropositivity and obesity in women, but not in men. On the other hand, in Lessan et al.’s study, an inverse association with gender was seen: Adv36-positive women were leaner than Adv36-negative women, whereas no such association was observed for men [21]. More research is needed in order to better understand the interplay between gender, obesity, and Adv36 infection.

Though we did not find any differences in serum lipid levels between the Adv36-seropositive and Adv36-seronegative groups, we did observe lower odds of being on lipid-lowering treatment for Adv36-positive participants, which was statistically significant even after adjustment for a variety of baseline characteristics, including age. This is consistent with animal studies that have shown an impact on lipid levels, although in those, the tendency was not always towards a less atherogenic profile. In human epidemiological studies, a favorable effect on serum lipid levels was reported in two reports: one from Na et al. (lower triglycerides regardless of BMI group, higher total cholesterol in the obese group, higher HDL cholesterol in the lean and obese group, lower total-cholesterol-to-HDL-cholesterol ratio [19]) and another from Sapunar et al. (lower VLDL cholesterol and triglycerides, no effect of Adv36-seropositivity on other cholesterol fractions, and no effect in the obese group [17]). On the other hand, many studies have shown no association between past Adv36 infection and plasma lipid levels [11,14,15,16,18,20,21]; some have even shown a deleterious association, such as Trovato et al. (decreased HDL cholesterol, increased triglycerides [10]) and Aldhoon-Hainerová et al. (increased total cholesterol, increased LDL cholesterol [26]), but these were unadjusted for BMI.

In Yamada et al.’s meta-analysis [22], Adv36-seropositivity was associated with a statistically significant but minor increase of LDL-cholesterol of 0.19 mmol/l, but there was no statistically significant impact of Adv36 infection on total cholesterol, HDL cholesterol, tryglicerides, glycaemia, or blood pressure, although this meta-analysis was conducted in 2012, and since then, quite a few new reports were published on this topic. Later studies have been included in Shang et al.’s [23], Xu et al.’s [24], and Marjani et al.’s [25] meta-analyses in 2014, 2015, and 2021, respectively, but only to assess obesity risk and not other components of the metabolic syndrome.

Similarly to our study, the vast majority of reports found no apparent effect on glucose homeostasis [10,11,15,16,20,21], with the exception of a study by Sapunar et al. [17], who found increased insulin sensitivity in Adv36(+) non-diabetics and lower fasting glycaemia in both Adv36(+) non-diabetic and diabetic patients compared to Adv36(−) subjects in a sample from Chile, and the study from Aldhoon-Hainerova et al. [26], who found lowered fasting glucose in Adv36(+) subjects. Reports on hypertension in this regard are scant and conflicting, with the report by Zhou et al. [20] showing lower systolic and diastolic blood pressure in Adv36(+) Han-Chinese subjects, and a report by Trovato et al. [10] showing an inverse relationship, with both reports unadjusted for confounding factors. Similar to our findings, other authors did not observe significant associations with regard to blood pressure [17,19].

In order to better characterize the cardiovascular risk in seropositive and negative patients, three additional non-traditional cardiovascular risk serum biomarkers were used: high-sensitivity C-reactive protein (hsCRP; linked to cardiovascular mortality risk [29]), resistin (also linked to cardiovascular mortality risk [30]), and calprotectin (linked to cardiovascular risk [31]). These three markers have been also linked to the pro-inflammatory state [32,33,34]. We are not aware of other reports of these markers according to Adv36-serostatus. In terms of these immune-associated cardiovascular-risk markers, we did not find any differences between Adv36(+) and Adv36(−) individuals, which strengthens the idea of benignity of past Adv36-infection in terms of long-term cardiovascular risk in PLHIV. Similarly, Karamese et al. [14] found no differences in terms of tumor necrosis factor alpha (TNF-α) or interleukin-6 (IL-6) serum concentrations when comparing Adv36(+) and Adv36(−) individuals from a general population. On the other hand, there is evidence from in vitro and animal studies to suggest that Adv36 infection increases macrophage infiltration into adipose tissue, as well as production of monocyte chemoattractant protein-1 (MCP-1), contributing to the pro-inflammatory state. This is further supported by the fact that serum MCP-1 levels were higher in Adv36(+) study participants than in Adv36(−) controls [35]. More research is needed to elucidate the interplay of Adv36 infection, immune activation (or the lack thereof), and cardiovascular risk.

## 5. Conclusions

We found no evidence to support the claim that past Adv36-infection is associated with an increased prevalence of cardiovascular risk factors or with elevated inflammatory markers. More research is needed to replicate these findings in other samples of PLHIV and to compare them with the HIV-negative population.

## Figures and Tables

**Table 1 viruses-14-01639-t001:** Laboratory and biomarker levels according to Adv36-serostatus—univariate analysis. Results of qualitative variables shown as percentages; results of quantitative variables shown as mean ± standard deviation.

Variable	Total	Adv36(+)	Adv36(−)	*p*
Percentage of sample	100%	26.4%	73.6%	-
Age [years]	44.0 ± 11.6	40.6 ± 11.0	45.3 ± 11.6	0.089
Male gender	92.3%	95.8%	91.0%	0.671
MSM	75.8%	87.5%	71.6%	0.167
Alcohol per week [g]	22.1 ± 58.6	19.7 ± 31.4	23.0 ± 65.8	0.810
Smoking [packyears]	5.5 ± 9.0	2.9 ± 5.2	6.5 ± 9.8	0.094
Moderate- or high-intensity physical activity per week [hours]	3.2 ± 3.6	2.5 ± 3.3	3.5 ± 3.7	0.256
History of IVDU	6.6%	4.2%	7.4%	1.000
History of drug use, any	26.4%	16.7%	29.9%	0.209
Time since HIV diagnosis [years]	8.1 ± 5.7	7.4 ± 6.6	8.4 ± 5.3	0.469
ARV treatment duration [years]	7.0 ± 4.9	5.6 ± 4.4	7.5 ± 5.1	0.128
CD4^+^ cell count nadir [cells/μL]	245 ± 232	261 ± 154	239 ± 255	0.717
CD4^+^ cell count [cells/μL]	529 ± 237	547 ± 172	525 ± 257	0.685
CD4^+^ cell percentage [%]	42.3 ± 13.6	46.5 ± 9.3	40.8 ± 14.3	0.073
CD8^+^ cell count [cells/μL]	732 ± 354	620 ± 204	772 ± 388	0.072
CD8^+^ cell percentage [%]	55.8 ± 12.6	52.0 ± 9.5	57.2 ± 13.4	0.086
CD4^+^/CD8^+^ ratio	0.86 ± 0.48	0.95 ± 0.33	0.82 ± 0.52	0.270
Receiving TAF	73.6%	83.3%	70.1%	0.208
Receiving an INSTI	75.8%	66.7%	79.1%	0.222
BMI [kg/m^2^]	25.6 ± 3.6	25.1 ± 3.0	25.8 ± 3.8	0.395
Waist circumference [cm]	93.9 ± 11.0	89.6 ± 7.7	95.5 ± 11.7	0.024
Hip circumference [cm]	102.9 ± 6.0	102.2 ± 4.3	103.1 ± 6.5	0.531
Waist-to-hip ratio	0.91 ± 0.08	0.88 ± 0.06	0.92 ± 0.09	0.014
On pressure-lowering treatment	26.4%	12.5%	31.3%	0.072
BP, systolic [mmHg]	133 ± 16	131 ± 12	134 ± 17	0.373
BP, diastolic [mmHg]	82 ± 11	81 ± 10	82 ± 11	0.525
On lipid-lowering treatment	28.6%	12.5%	34.3%	0.042
Cholesterol, total [mmol/L]	5.09 ± 1.17	5.10 ± 1.16	5.09 ± 1.18	0.967
Cholesterol, LDL [mmol/L]	3.06 ± 0.95	3.02 ± 0.92	3.07 ± 0.97	0.825
Cholesterol, HDL [mmol/L]	1.29 ± 0.39	1.30 ± 0.50	1.29 ± 0.34	0.892
Triglycerides [mmol/L]	1.54 ± 0.81	1.41 ± 0.62	1.59 ± 0.87	0.371
On glucose-lowering treatment	5.5%	4.2%	6.0%	0.687
Serum glucose, fasting [mmol/L]	5.36 ± 0.72	5.41 ± 0.74	5.34 ± 0.72	0.673
Serum insulin, fasting [mU/L]	15.2 ± 19.4	15.6 ± 31.7	15.1 ± 13.5	0.912
HOMA-IR	3.78 ± 5.12	3.91 ± 8.24	3.74 ± 3.64	0.899
hsCRP [mg/L]	2.60 ± 3.97	2.12 ± 3.52	2.89 ± 4.21	0.431
Calprotectin [ng/mL]	6.72 ± 2.07	6.68 ± 1.63	6.74 ± 2.22	0.911
Resistin [ng/mL]	11.1 ± 8.58	10.7 ± 6.5	11.3 ± 9.3	0.754

Adv36—Adenovirus 36; MSM—men who have sex with men; IVDU—intravenous drug use; ARV—antiretroviral; TAF—tenofovir alafenamide fumarate; INSTI—integrase-strand inhibitor; BMI—body mass index; BP—blood pressure; HOMA-IR—homeostatic model assessment—insulin resistance; LDL—low-density lipoprotein; HDL—high-density lipoprotein; hsCRP—high-sensitivity C-reactive protein.

**Table 2 viruses-14-01639-t002:** Adv36-serostatus and anthropometric measurements adjusted through multiple regression analysis by: age, gender, sexuality (MSM vs. other), ARV therapy (receiving TAF vs. not receiving TAF; receiving an INSTI vs. not receiving an INSTI), and CD4^+^ percentage.

Dependent Variable	Ad36(+) Patients (in Comparison to Ad36(−) Patients)	*p*
BMI	−0.5 kg/m^2^	0.579
Waist circumference	−4.5 cm	0.086
Hip circumference	+0.08cm	0.135
Waist-to-hip ratio	−0.03	0.102

Adv36—Adenovirus 36; MSM—men who have sex with men; ARV—antiretroviral; TAF—tenofovir alafenamide fumarate; INSTI—integrase-strand inhibitor; BMI—body mass index.

**Table 3 viruses-14-01639-t003:** Adv36-serostatus and cardiovascular risk factors adjusted through multiple regression analysis: age, gender, sexuality (MSM vs. other), ARV therapy (receiving TAF vs. not receiving TAF; receiving an INSTI vs. not receiving an INSTI), CD4^+^ percentage, treatment (where appropriate, see indexes), and waist-to-hip ratio.

Dependent Variable	Ad36(+) Patients (in Comparison to Ad36(−) Patients)	*p*
BP, systolic ^1^	+0.3 mmHg	0.931
BP, diastolic ^1^	+0.6 mmHg	0.801
Total cholesterol ^2^	+0.1 mmol/L	0.665
LDL cholesterol ^2^	+0.0 mmol/L	0.944
HDL cholesterol ^2^	−0.0 mmol/L	0.910
Triglyceride ^2^	−0.1 mmol/L	0.654
Fasting glucose ^3^	+0.3 mmol/L	0.129
Fasting insulin ^3^	+3.6 mU/L	0.500
HOMA-IR ^3^	+1.2	0.408
hsCRP	−1.5 mg/L	0.193
Calprotectin	+0.0 ng/mL	0.969
Resistin	−0.2 ng/mL	0.930

^1^—adjusted for receiving pressure-lowering treatment; ^2^—adjusted for receiving lipid-lowering treatment; ^3^—adjusted for receiving glucose-lowering treatment. Adv36—Adenovirus 36; MSM—men who have sex with men; ARV—antiretroviral; TAF—tenofovir alafenamide fumarate; INSTI—integrase-strand inhibitor; BP—blood pressure; LDL—low-density lipoprotein; HDL—high-density lipoprotein; HOMA-IR—Homeostatic Model Assessment—Insulin Resistance; hsCRP—high-sensitivity C-reactive protein.

**Table 4 viruses-14-01639-t004:** Adv36-serostatus and odds of being on particular treatment adjusted through multiple logistic regression by: age, gender, sexuality (men who have sex with men vs. other), ARV (receiving TAF vs. not receiving TAF; receiving an INSTI vs. not receiving an INSTI), CD4^+^ cell percentage, and waist-to-hip ratio.

Treatment	Odds Ratio of Ad36(+) Patients in Comparison to Ad36(−) Patients	Odds Ratio 95% Confidence Interval
Pressure-lowering treatment (receiving vs. not receiving)	0.40	0.07–1.78
Lipid-lowering treatment (receiving vs. not receiving)	0.23	0.04–0.91
Glucose-lowering treatment (receiving vs. not receiving)	Not calculated due to low number of participants on glucose lowering treatment	

Adv36—Adenovirus 36; MSM—men who have sex with men; ARV—antiretroviral; TAF—tenofovir alafenamide fumarate; INSTI—integrase-strand inhibitor; BP—blood pressure; LDL—low-density lipoprotein; HDL—high-density lipoprotein; HOMA-IR—homeostaric model assessment—insulin resistance; hsCRP—high-sensitivity C-reactive protein.

## Data Availability

Data available upon reasonable request.

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
