# Peer review of "Adenovirus 36 Infection in People Living with HIV—An Epidemiological Study of Seroprevalence and Associations with Cardiovascular Risk Factors"

_viruses, 2022, doi:10.3390/v14081639_

Round 1
Reviewer 1 Report
I thank the authors for revising the manuscript according to the reviewers' suggestions.
Reviewer 2 Report
The authors have answered reviewers comments.
This manuscript is a resubmission of an earlier submission. The following is a list of the peer review reports and author responses from that submission.
Round 1
Reviewer 1 Report
Sapula et al propose a cross sectional biological study of Adenovirus 36 serology in PLWH, and explore the correlation with different cardiovascular risk factors including obesity which has been linked in several study to Adv36 seropositivity. While this specific study in PLWH is important data to describe, the conclusions are not supported by the results.
Major remarks:
- Male gender is largely predominant in the cohort (92%), but the link between obesity and Adv36 has been shown mainly in women in princeps articles (Atkinson et al, Trovato et al)
- The technique for Adv 36 serology is not described. It is an important issue in the study as serum neutralisation assay is the gold standard for determining the serology.
- The inclusion criteria and the objectives of the study are not clearly explained.
- Pressure, lipid and glucose -lowering treatment association with Avd36 seropositivity are not end points of the study
- The conclusion should not support a benign effect of Adv36 on cardiovascular risk factors in PLWH including because of the relatively low number of patients included
Minor remarks:
- The institution giving the ethical consent for the study should be quoted.
- P values for waist circumference and waist to hip ratio are not significant with regard to the number of tests performed.
Reviewer 2 Report
In the current research article by Mariusz et al., the authors have studied the correlation of Adv36 seropositivity with obesity/Cardiovascular risk factors in people living with HIV (PLHIV). In their cross-sectional study of 91 PLHIV with about 26% Adv36 seropositive individuals, they found no positive in fact, slightly negative correlation of Adv36 infection with obesity in PLHIV. In general, previous research studying the correlation of Adv36 infection with obesity has led to very controversial results indicating that there could be multiple genetic/epigenetic factors affecting this correlation confounding the outcome. Even in the current study with more the 90% males there is a lack equal distribution with respect to gender. While all the subjects belong to PLHIV cohort and hence on ART therapy, it is not fully known if ART drugs would affect cardiovascular health differently based on individuals’ genetic make-up or epigenetic factors. Based on these factors, it is difficult to conclude that the negative correlation observed in the study is true. In my opinion, the study design could be improved by keeping equivalent number of males and females to rule out the gender related effects on correlation that could skew the results. Furthermore, more participants with Adv36 seropositivity could have led to more confidence in the results. Overall, I appreciate the authors for studying this correlation in PLHIV, however, would have liked them to plan the study very carefully learning from the paradoxical results obtained from similar numerous studies done in the past.